# New Patterns for Highly Pathogenic Avian Influenza and Adjustment of Prevention, Control and Surveillance Strategies: The Example of France

**DOI:** 10.3390/v16010101

**Published:** 2024-01-10

**Authors:** Axelle Scoizec, Eric Niqueux, Audrey Schmitz, Béatrice Grasland, Loïc Palumbo, Adeline Huneau-Salaün, Sophie Le Bouquin

**Affiliations:** 1Ploufragan-Plouzané-Niort Laboratory, Epidemiology Health and Welfare Unit, French Agency for Food, Environmental and Occupational Health and Safety (ANSES), BP53, 22440 Ploufragan, France; adeline.huneau@anses.fr (A.H.-S.); sophie.lebouquin-leneveu@anses.fr (S.L.B.); 2Ploufragan-Plouzané-Niort Laboratory, Avian & Rabbit Virology, Immunology & Parasitology Unit, French Agency for Food, Environmental and Occupational Health and Safety (ANSES), BP53, 22440 Ploufragan, France; eric.niqueux@anses.fr (E.N.); audrey.schmitz@anses.fr (A.S.); beatrice.grasland@anses.fr (B.G.); 3Research and Scientific Support Department (DRAS), Wildlife Health and Agricultural Ecosystem Functioning Department (SantéAgri), National Biodiversity Office (OFB), 9 Av. Buffon, 45100 Orléans, France; loic.palumbo@ofb.gouv.fr

**Keywords:** avian influenza, HPAI, zoonotic risk, enzootic, prevention and control, poultry, wildlife

## Abstract

From 2020 up to summer 2023, there was a substantial change in the situation concerning the high pathogenic avian influenza (HPAI) virus in Europe. This change concerned mainly virus circulation within wildlife, both in wild birds and wild mammals. It involved the seasonality of HPAI detections, the species affected, excess mortality events, and the apparent increased level of contamination in wild birds. The knock-on effect concerned new impacts and challenges for the poultry sector, which is affected by repeated annual waves of HPAI arriving with wild migratory birds and by risks due to viral circulation within resident wild birds across the year. Indeed, exceeding expectations, new poultry sectors and production areas have been affected during the recent HPAI seasons in France. The HPAI virus strains involved also generate considerable concern about human health because of enhanced risks of species barrier crossing. In this article, we present these changes in detail, along with the required adjustment of prevention, control, and surveillance strategies, focusing specifically on the situation in France.

## 1. Introduction

Since the emergence of the high pathogenicity avian influenza (HPAI) A/Goose/Guangdong/1/1996 (H5) virus lineage in Asia, first detected in 1996 and responsible for initial epizootic expansion in China and Southeast Asia from 2003 onwards, intercontinental HPAI panzootics of this viral lineage have followed one another since 2005. The first panzootic wave of zoonotic HPAI clade 2.2 (H5N1) in 2005–2006 reached Asia, Europe, the Middle East, and West Africa [1]. The second, in 2009–2010, of non-zoonotic clade 2.3.2.1c (H5N1), affected Asia and Eastern Europe without any reports of contemporary human infections [2]. The next wave in 2014–2015 involved the spread of two lineages, with no reports of zoonotic transmissions either: once again, the non-zoonotic clade 2.3.2.1c (H5N1) in Asia, the Middle East, East Africa, and Cameroon, and the clade 2.3.4.4 group A (H5N8, N2, N1) in East Asia, Korea, Japan, Southeast Asia, Europe, and North America [3,4]. The emergence of clade 2.3.4.4b in China in 2008 [5] then gave rise to the first panzootic wave in 2016–2017 (HPAI H5N8, N5, N6, etc.) affecting Asia, the Middle East, Europe, West, Central, East, and South Africa. A second wave then began in 2020 which was to be followed by a clear rise in intensity with a succession of seasonal epizootics in Europe, Africa, and Asia, extending subsequently to North and South America. During these different panzootic waves, Europe was affected to highly variable extents [6], as shown in Figure 1. This ranged from a few sporadic detections in wild birds and poultry farms to massive detections in wild birds and, in parallel, major disseminations within the poultry industry, particularly on duck and goose farms (especially in France [7] and Hungary [8]), but also on turkey farms (e.g., in Germany [9]). The general pattern of introduction and extension of these epizootics in Europe has, until now, involved first, the arrival of new HPAI viral strains in the autumn via infected migratory wild birds, in particular wild birds of the family Anatidae, which arrive from their nesting and moulting sites in arctic and subarctic zones. This was followed by detections in migratory or sedentary wild birds that were generally sporadic. Sporadic introductions to poultry farms could then lead to secondary, sometimes major, farm-to-farm spread. The amplification observed since 2020–2021 has introduced substantial changes in this pattern. Changes are taking place both in the circulation of this 2.3.4.4b lineage within wild fauna in Europe and around the world, and in its impact on the poultry industry. In this article, we discuss the new situation concerning the ecology and epidemiology of this disease in Europe, and more specifically in France. We focus on both wild birds and poultry farms, and on human health. We also discuss the pragmatic adjustments made to deal with this new situation.

## 2. The New Face of HPAI in Wild Birds in France and Europe

### 2.1. Very High Viral Circulation in Wild Birds in Europe

As shown in Figure 2, the number of HPAI H5 virus detections among wild waterfowl was very high in each of the three postnuptial fall migration seasons since winter 2020–2021, revealing a very high level of circulation of these viruses in this compartment. The level did not appear to diminish from one year to the next during this period [5]. The situation differs from that encountered during the previous panzootics. Previously, in comparison with the autumn/winter period, marked by a significant wave of detections in wild birds, there was a collapse or absence of detections in wild birds during the same period of the following year (Figure 3). This indicates that, in epizootic waves prior to 2020, viral circulation in migratory populations did not appear to remain at a high level from one year to the next.

At the same time, since summer 2021, HPAI H5 viruses have been circulating in wild birds in Europe, even during warmer months, i.e., in late spring and summer. This represents a real change: previously the highly seasonal nature of these epizootics, linked to the movements of wild bird reservoirs (migratory wild birds of the family Anatidae, and possibly other migratory families) and summer weather conditions unfavorable to the survival of HPAI viruses (HPAIV) in outdoor environments, meant that there was no detection at this time of year. Indeed, in France, the moment of arrival of migratory waterfowl after their nesting and moulting periods, is mainly from September to December, depending on the species [12,13]. Nesting and moulting are risk periods when significant contamination of these populations may occur due to: (i) close contact between numerous bird populations of various species arriving from many parts of the world—Asia, Europe, Africa—on shared arctic or sub-arctic nesting and moulting grounds [4,14]; (ii) juvenile birds with a naïve status representing an important proportion of the nesting population, in contact with adults who have potentially exchanged numerous viruses during the moulting and reproduction periods, and this would support the spread of these viruses through infected bird populations during their fall migrations [15].

In 2021, in Europe, detections in wild birds during the summer period mainly concerned the family Anatidae, albeit in relatively small numbers. These detections were due predominantly to subtype A(H5N1) virus infections that were two-fold more frequent than A(H5N8) cases in wild birds during that summer period, in contrast with about 10-fold less A(H5N1) than A(H5N8) cases in wild birds in the whole 2020–2021 epidemic period [16]. The virus genotypes concerned were mainly the H5N8-Genotype A Duck/Chelyabinsk-like, the H5N1-Genotype C Eurasian wigeon/Netherlands-like and the H5N1-Genotype AB Duck/Saratov-like [17]. The following summer period in 2022 saw an explosion in detections among colony-nesting seabirds, mainly families Laridae and Sulidae, with respectively, H5N1-Genotype BB Herring gull/France-like and H5N1-Genotype C Eurasian wigeon/Netherlands-like viruses, and also, to a lesser extent, among the family Anatidae as well as raptors and other species [5]. These cases among the Anatidae family were mainly due to H5N1-Genotype AB Duck/Saratov-like viruses. In 2023, massive circulation began in January among Laridae populations (mainly gulls and terns) in their winter staging areas, then from mid-April onwards in nesting areas all over Europe, extending to all colonial *Laridae* species. This suggests that there may be endemization of HPAI within indigenous wild bird populations in Europe, and even in France. This concerned principally the viruses of the H5N1-Genotype BB.

Detections in wild birds have brought to light viral circulations in species that were not or only slightly affected until now. This was the case in Europe in populations of Scolopacidae (season 2020–2021), Sulidae (season 2021–2022) and Laridae (seasons 2021–2022 and 2022–2023), as well as locally in France and Spain in Eurasian Griffon vultures (*Gyps fulvus*) [5,18,19,20,21](see Figure 4). A new spectrum of heavily affected species has now been observed.

At the same time, in certain areas of France, numerous detections of the virus on poultry farms or in backyard flocks (following phylogenetic analyses) have been attributed with high likelihood to introductions from wild birds, without any direct identification of the wild source by the surveillance system. It is therefore suspected that significant viral circulation in wild birds is occurring without mortality in certain populations, particularly among wild *Anatidae* species, and that a part of the viral circulation is going undetected or under-detected by our current event-based mortality surveillance system.

In brief, the years 2005–2019 in Europe were marked by the irregular arrival of panzootic waves of HPAI in wild birds during their postnuptial fall migrations. In contrast to this previous situation, we are currently facing a considerable rising tide of virus circulation seemingly maintained at a very high level in migratory bird populations from one year to the next, and a probable endemization in resident wild bird populations. These changes have led to strong waves of infection every year since the fall of 2020 and viral circulation in wild birds in Europe all year long.

### 2.2. Major Episodes of Excess Mortality within Wild Birds

The 2020–2021, 2021–2022, and 2022–2023 seasons were marked by major HPAI-related mass mortality episodes among several unexpected wild bird populations: red knots (*Calidris canutus*), *Laridae*, *Sulidae*, and Griffon vultures. These excess mortalities, on a European or even global scale, can represent a real threat to the conservation of endangered species that have been affected, such as Griffon vultures, gannets, and certain species of terns. For example, in France, the only northern gannet (*Morus bassanus*) nesting colony located at the “Sept-Îles” nature reserve (Côtes d’Armor, Brittany) was severely affected in 2022, with an estimated 80% loss of chicks and very high mortality of adult birds. More than 1200 dead gannets were counted in areas of active viral circulation in Brittany between July and September 2022 [23]. Many other northern gannet colonies in Europe have been affected (e.g., Scotland, Wadden Sea), as well as in North America (e.g., Canada) [24].

These episodes of mass excess mortality of wild birds also represent a public health risk. In areas of high human use, the general public has been confronted with the presence of sick, dying, and dead animals. Such areas included coastal areas in summer, and urban or suburban areas such as around seagull wintering staging areas in Île-de-France, a very densely populated region. This resulted in increased exposure of the general public to HPAIV-infected birds, and a challenge in coping with the inflow of sick and dead birds over large areas.

### 2.3. Adjusting HPAI Surveillance in Wild Birds

HPAIV surveillance in wild birds in France is built on an event-based surveillance system of birds that are found dead or moribund. Surveillance is carried out via the National Network for Wildlife Disease Surveillance (SAGIR), which is a network of field operators, animal health specialists, and specialized laboratories [25,26], in accordance with the official instructions of the French Ministry of Agriculture and Food Safety [27,28]. For HPAI surveillance, this system is focused on the collection of target species and uses thresholds for triggering sampling and analysis depending on the number of dead birds at the place of discovery, the species concerned, the area category (e.g., wetlands used by migratory birds), and the regulatory level of HPAI risk linked to wild birds. The target species correspond to the list of wild bird species targeted for passive surveillance of H5 HPAI viruses in the European Union [29]. The bird families targeted are Anatidae, Laridae, Accipitridae, Falconidae, Strigidae, Rallidae, Ardeidae, Phalacrocoracidae, Ciconiidae, Scolopacidae, Pelecanidae, Podicipedidae and Gruidae. For example in France, whatever the HPAI risk level and the area category are, a discovered dead swan will be always sampled for avian influenza surveillance as these species are considered the sentinel of the disease. For other *Anatidae* species, when the risk level is considered low, it is only when the threshold of three dead birds is reached that sampling will be performed for avian influenza surveillance whereas a single dead bird is sufficient in higher risk situations. The new patterns of HPAI virus circulation among wild birds in Europe have then requested adjustments to this surveillance protocol.

First, the system was not designed for episodes of mass excess mortality lasting several weeks in wild bird populations. It was not tailored to track the dynamics of these excess mortalities, and its strict application would have entailed a multitude of samples and analyses of limited value for tracking viral strains, and identifying infected areas and species while overloading and destabilizing the whole surveillance scheme. The French Biodiversity Agency (OFB), in charge of coordinating this surveillance system, quickly implemented reactive adjustments after discussions with the French competent authorities and scientific institutes. Subsequently, recommendations for adjusting surveillance to the new context were put forward by a dedicated working group of the French Platform for Epidemiological Eurveillance in Animal Health (ESA Platform) [30,31].

The OFB first set up a system for monitoring the dynamic of the episodes of excess mortality via its network of SAGIR correspondents: the aim of this system is not to count mortalities precisely but to assess their dynamics by species and by area, based on the recording of all reports received via field observers.

Second, the surveillance system has been reviewed, by revising the priority species targeted for analysis, and by adjusting the protocol to episodes of excess mortality, in order to reduce sampling while monitoring viral strains over time, to ensure that excess mortality is always linked to HPAI and identify as many species affected by this viral circulation as possible. The OFB also ensures that the appropriate management measures are properly implemented in accordance with national regulations.

An emerging concern is the probable circulation of HPAI viruses within wild bird populations undetected by the event-based surveillance system. In order to address this new surveillance challenge, the dedicated group in the ESA Platform worked on proposing a new active surveillance program. Although this kind of approach was previously considered to be less cost-effective [32] and was abandoned in France in 2012 following the recommendations of the EU Commission [33], it must be reconsidered within this new context [34,35,36]. To this end, the group proposed to start with an initial proof-of-concept study including the active surveillance of hunted birds (at least for *Anatidae* species) and birds trapped for ringing. The aim of this study would be to develop the tools needed to identify viral circulation within these populations in the French context. The ultimate goal of the program would be to determine the epidemiologic role of these populations enabling this undetected circulation (potential reservoirs, relay hosts within the avifauna and towards the domestic compartment, etc.), and to describe the dynamics of virus circulations within them (periods, areas, etc.).

### 2.4. Managing Dead Wild Birds: A New Challenge

Public Services have had to adjust to the management of a large number of contaminated dead birds in areas heavily frequented by the general public. Collection was carried out by the municipal services of the affected communes, and the dead birds were then sent to rendering services, as stipulated in the French Rural and Maritime Fishing Code (https://www.legifrance.gouv.fr/codes/section_lc/LEGITEXT000006071367/LEGISCTA000006168834/2020-11-06/, accessed on 15 September 2023). As a result, urgent training courses for municipal staff were organized by various public organizations. Difficulties were encountered with the collection of carcasses by the rendering services. Based on the conditions for the collection of domestic animal carcasses, these services only collected carcasses with a minimum total cumulative weight of 40 kg, but this threshold was rarely reached (for example, a tern weighs a few hundred grams) on each day of carcass collection, resulting in delays of several days before the rendering service arrived. Furthermore, this posed the problem of the storage of decomposing carcasses, as the communes were not equipped with freezers for this one-off service.

It is, therefore, necessary to design and deploy a standardized training system for municipal employees potentially in charge of the collection of wild dead animals potentially infected with zoonotic and infectious pathogens. It is also important to review the collection criteria or system with rendering companies, in order to comply with the compulsory two-clear-day deadline, without weight criteria, for the removal of animals found dead without a known owner (such as wild animals). Of course, as with the collection of carcasses in livestock outbreaks of avian influenza, rendering companies must be systematically informed of the infectious risk, for the purpose of adjusting their operations to it (e.g., collection at the end of rounds, protection of drivers, and reinforced sealing of collection skips).

### 2.5. Adjusting the HPAI Wildlife Risk Level System

In France, the ministerial order of 16 March 2016 set up the system of HPAI risk levels related to wild birds. The system has three risk levels: negligible, moderate, or high. These levels were defined according to the detection of HPAI within wild birds and especially within migratory species (location, dynamics, and period of the year). The French Agency for Food, Environmental, and Occupational Health and Safety (ANSES) proposed a revision of the system [37] in response to the changes in virus circulation within wild birds. This proposal distinguishes two risk scales for the combined evaluation of the risk of infection by HPAI viruses in the poultry farm compartment: first, a scale in the context of wild bird migrations, and second, another in the context of infection within resident wild birds. “Resident” wild birds are defined here as sedentary birds and migratory birds staying several weeks in France. The proposed system takes into account the type of zone where the risk applies (i.e., wetland area, high-density poultry area), the time of year, and the biotope of infected resident birds (strength of the link with the poultry farming compartment).

### 2.6. Protecting Endangered Species: Conservation Issues

In response to the major damage caused by HPAI to several endangered bird populations, we need to find ways of protecting these small-scales-threatened populations. In this context, the implementation of vaccine protocols for endangered species could be considered and developed. Currently, a vaccine trial for the protection of the Amsterdam albatross (*Diomedea amsterdamensis*) in the Southern and Antarctic Territories is being discussed in conjunction with the French national action plan for this species [38]. Other populations such as gannets and certain tern species could also be considered for such an approach. To do this, methods suitable for each species would be needed, to carry out vaccination and monitor results. Interestingly, this type of approach is currently implemented in the United States to protect Californian condors threatened by HPAI viruses [39,40].

## 3. New Impacts and Challenges for Poultry Farms

### 3.1. Evolving HPAI Risks for Poultry Farms

Poultry farms now face an HPAI risk from migratory and resident wild birds in all seasons of the year. While the risk remains much higher in the winter and shoulder seasons, sporadic cases of introduction into domestic fauna (poultry farms and backyard flocks) have also been observed outside these periods (see Figure 5). However, keeping the poultry indoors [41,42], which is one of the main measures to prevent introduction and spread on poultry farms, is difficult to apply in summer, both from a zootechnical and animal welfare point of view.

Furthermore, the risk is increasing in new areas and new sectors, potentially related to greater viral circulation in wild birds. The number of primary introductions from wild bird sources into poultry farms has increased with each season since 2020–2021, leading to epizootics through farm-to-farm spread to previously little-affected areas [43,44]. Examples of this extension of the areas affected by farm-to-farm epizootics are the 2021–2022 and 2022–2023 seasons in the Pays de la Loire region of western France, and the 2021–2022 season in the Center-Western part of France (see Figure 6). During the 2022–2023 season, there were also many primary introduction cases in Brittany, followed by an epizootic of 20 outbreaks, mainly affecting poultry farms of the egg-laying sector in the Côtes d’Armor department. This epizootic in Brittany highlighted the difficulties in outbreak management and the risks of spread in this specific sector. This is, for example, due to the very large size of some laying hen farms, the management of poultry droppings (large volumes, risk of spread due to the powdery nature of the material), and the difficulties in tracing egg collection rounds and organizing them in accordance with the restricted zones around HPAI outbreaks.

For several recent seasons, we have seen that in areas with high poultry densities, and in particular high duck densities, viruses can overwhelm our prevention and control measures, leading to major epizootics involving several hundred outbreaks [41,45]. This has been the case on several occasions in southwestern France, but also in the Pays de la Loire region, despite constant improvements in biosecurity measures in the various links of the poultry supply chains. Each season, the viruses seem to outstrip our adjustments: they appear to be more contagious, exhibit changes in clinical expression that can delay early detection, reach new areas and new sectors, etc. In this context, despite the constant adjustments made, it is rather difficult to accurately anticipate new crises.

**Figure 6 viruses-16-00101-f006:**
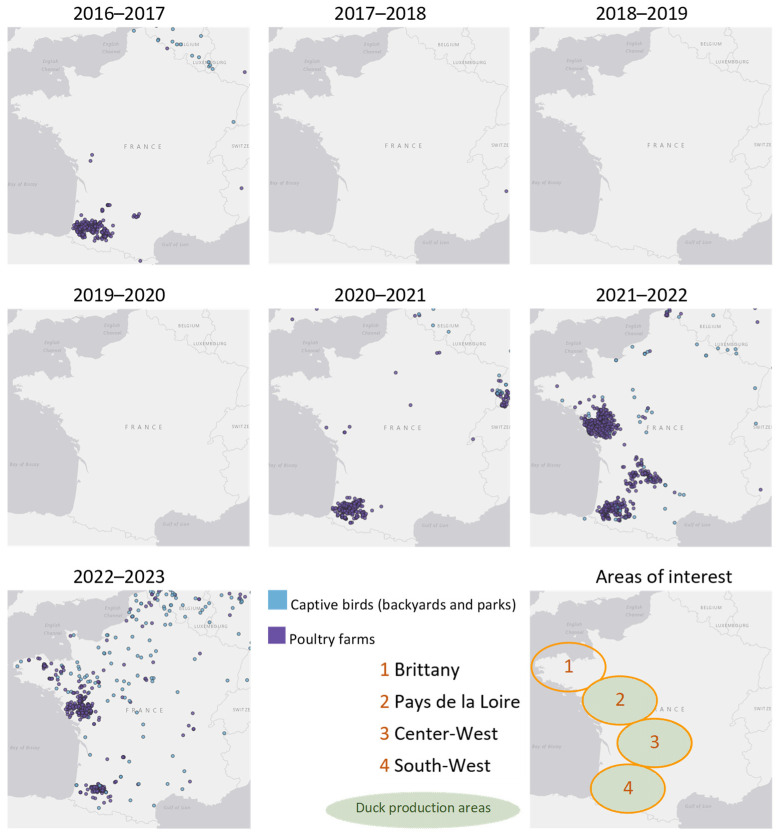
Geographical distribution of the detections of HPAI H5 virus on poultry farms and in captive birds in France over annual HPAI seasons (one annual HPAI season is from August to July) since the first HPAI H5 clade 2.3.4.4b strain detection in 2016 up to July 2023 (Source: ESA Platform, French Animal Health Epidemic Intelligence [46]).

### 3.2. Need to Adjust Surveillance, Management, and Prevention Measures

In order to meet the challenge of early detection of any introduction from wild bird sources into the poultry farm compartment, and to limit the risk of spread within the compartment as quickly as possible, the ESA Platform’s Avian Influenza group has revised the event-based surveillance system for poultry farms. A protocol for enhanced active surveillance in areas at high risk of epizootic disease in poultry farms has been proposed in addition to the extant annual active surveillance and regulatory surveillance around detected outbreaks.

For the event-based surveillance system, the clinical signs expected with circulating strains or strains likely to circulate have been amended, so that the system takes into account changes in clinical signs depending on viral strains and infected species. A testing-for-exclusion laboratory diagnosis is now possible, without triggering official suspicion status, which is always burdensome for farmers and production organizations. This measure was added to increase the surveillance system’s sensitivity by including the investigation of clinical cases deemed unlikely because they could be attributed to other diseases. The revision has also highlighted the need to prioritize the removal of dead or sick animals from suspect batches.

The protocol for active enhanced surveillance of poultry farms takes into account the context in which it is implemented: (i) the level of HPAI risk (related to infected wild birds or to detections on poultry farms), (ii) areas at risk of spread (based on poultry density levels), and (iii) species targeted for current strains (i.e., ducks and geese, due to the potential lack of clinical signs of infection or the duration of the preclinical excretion period in these species). Like in event-based surveillance, these criteria will be revised in line with changes in clinical signs according to viral strain/bird species pairs. Surveillance methods are also evolving, with weekly swabbing of the environment of poultry houses, and with swabbing of a representative sample of animals before any animal transfer to another farm. In addition, professionals and local authorities were able to adjust these protocols to suit the given local context. For example, during the HPAI crisis of winter 2022–2023 in the Vendée department, Pays de la Loire, monitoring by environmental swabbing became a twice-weekly procedure to improve early warning.

The measure involving mandatory confinement of poultry, which is used to prevent introduction to poultry farms and farm-to-farm spread, has been revised by the ANSES dedicated expert group [47] in order to adjust it not only to the HPAI risk but also to both climate conditions and animal welfare. Following a risk analysis, new rules for broilers and other galliforms were proposed to mitigate their confinement, except for turkeys and laying hens for which total confinement during the mandatory periods remains the rule. Moreover, the group’s work emphasized the need for studies to evaluate alternative methods of sheltering.

Furthermore, innovative actions are required to prepare for the new management challenges posed by the size and number of infected farms. We must consider all possible depopulation systems for poultry batches, with regard to the species, type of building, batch size, etc. This type of approach must also be adopted for the means and methods of dead bird disposal. The aim will be to provide a toolbox enabling methods and resources to be adjusted to each case. In the field, French authorities have already diversified culling methods, for example using in-building gassing when possible, rather than gassing in containers during the major epizootic on poultry farms in Pays de la Loire in 2021–2022. They have also implemented the burial of carcasses on farms and in landfill sites when rendering plant capacities were exceeded [48].

To meet these new challenges, we need to be pragmatic, break with dogma, and be creative in our search for new solutions. It is also clearly necessary to review the general model, which relies mainly on state resources and public procurement to provide the necessary means. The industry should be a driving force behind proposals and a full-fledged stakeholder in the search for and implementation of solutions, as was the case during the mobilization of slaughterhouses and the creation of a platform for preventive depopulation in southwestern France during previous HPAI crises. Over and above these aspects, the issue of avian influenza must be taken into account in the design, sizing, and siting of farms by professionals. This concerns farm protection against the risk of AI, management in the event of infection (means provided by the profession to enable rapid culling for very large farms), and ease of site decontamination (buildings, equipment, runs, etc.). This is particularly important for large, indoor aviary and cage layer farms, where processes for equipment cleaning and effluent management (dry droppings) are very complex. The approach should be at the single farm level, but also by production area.

### 3.3. Adapting Poultry Sector

Reducing density, particularly of ducks, in areas with very high densities, is a measure that has been identified as a potential major improvement by ANSES’s expert appraisal work [49], as well as by modeling research [50]. During the 2022–2023 season, voluntary or involuntary reductions in density appear to have had a major effect on the size of epizootics. The number of full duck production units between December 15 and January 15 decreased very significantly by pausing the restocking of poultry houses with ducklings in 68 communes in the southwest area. These communes are located in high-risk zones (i.e., high poultry density, wet areas) and had been systematically contaminated during previous crises. The “Adour Plan” [51], set up by and on the initiative of industry, was most likely an effective, protective tool against an HPAI epizootic last winter in the southwest of the country: 331 outbreaks were detected in poultry during winter 2021–2022 vs. 20 outbreaks during winter 2022–2023.

In the Pays de la Loire region, lower duck densities following the shortage of batches of one-day-old ducklings after the 2021–2022 epizootic, which had affected the majority of breeding duck farms, are very probably one of the main reasons why the epizootic was limited in size. This occurred in the context of high viral circulation in wild birds, with over 800 outbreaks in 2021–2022 vs. around 230 in 2022–2023.

At the same time, we can see that the epizootic risk remains very high, even in the shoulder season (spring) in areas with high densities of poultry farms, and more particularly of duck farms, in the event of a primary outbreak. This was illustrated by the “surprise” epizootic in the southwest from May to early June 2023, with 85 outbreaks detected, in an area with a high density of duck farms for foie gras production. Many of these duck farms had re-opened their free-range outdoor access after official authorization was granted at the end of April, and their production units were at their maximum capacity to compensate for the production shortage linked to the Adour Plan.

As highlighted in ANSES’s expert assessments [49], reducing the number of poultry establishments in the highest poultry density areas and more specifically the highest duck density areas in high-risk periods is an important measure to reduce the epizootic risk. The challenges ahead in order to achieve this are numerous: (i) avoid recreating hyper-dense areas elsewhere in the country, (ii) separate areas where strategic farms are located (such as breeding farms for poultry selection) from poultry production areas and from wetlands areas, and (iii) manage these reductions in density over several seasons of the year.

### 3.4. Vaccination: A Paradigm Shift

Repeated HPAI epizootics in France and Europe caused huge economic losses for industries and public finances alike because of outbreak management measures (culling, compulsory fallowing, etc.) and production losses. In addition, regaining disease-free status at the national level was difficult because of sporadic HPAI outbreaks detected in domestic poultry in all seasons, entailing almost constant barriers to export to many countries. Vaccination has, therefore, become an option considered by France, and more widely at the European level.

However, there is very little data on existing vaccines or vaccines in development for certain species of major interest for HPAI control in Europe, such as palmipeds, turkeys, or laying hens. Importantly, the prior existence of a market drives the search for vaccine solutions and vaccination was not previously authorized for poultry farms in Europe. Dealing with this situation and as support to the European Food Safety Authority (EFSA) for its scientific opinion on vaccination against highly pathogenic avian influenza [52], a study was set up in several European countries, starting in 2022, to assess the efficacy of vaccination in different poultry species: in France for mulard ducks, in Italy for turkeys, in the Netherlands for laying hens, and in Hungary and the Czech Republic for geese. Efficacy is assessed in four of its features: clinical protection, reduction in virus excretion, reduction in virus dissemination, and the ability to differentiate between vaccinated and infected animals (DIVA). Initial results from the mulard duck study indicate that vaccines can be highly effective in limiting transmission [53,54]. The challenges were performed with a strain representative of the recent HPAI viruses circulating in the field (A/chicken/France/D2107428/2021 (H5N1) clade 2.3.4.4b) that was heterologous compared with the H5 sequences of the two evaluated vaccines (derived from A/duck/France/161108h/2016 (H5N8) clade 2.3.4.4b and A/duck/China/E319-2/2003 (H5N1) clade 2.3.2 viruses).

In parallel with this study, a prospective work on several possible vaccination scenarios was carried out by ANSES and the results have been published [55].

Surveillance of vaccinated poultry flocks is essential to avoid silent circulation of the pathogenic virus and for early detection of an emerging virus not covered by vaccination. The circulation of low pathogenic avian influenza viruses in domestic poultry populations can lead to the emergence of highly pathogenic strains via viral mutation phenomena, as in domestic poultry flocks in France in 2015–2016 [56]. Delegated Regulation 2023/361 (Annex XIII) of the Official Journal of the European Union [57] lays down the common basis for the implementation of vaccination and the surveillance of vaccinated poultry between EU Member States.

Recent epizootics on poultry farms in France have shown that prevention and control measures are not always sufficient to control farm-to-farm spread of HPAI viruses, particularly in areas with high duck densities and in seasons that are favorable or very favorable for the maintenance of infectious viruses in the environment. Vaccination would increase the ability of implemented biosecurity and surveillance measures to control the spread of these strains within the poultry sector if it sufficiently reduces the capacity of these HPAI virus strains to spread from contaminated vaccinated farms. Anyway, vaccination is not an easy measure to implement for regulated diseases. There are difficulties in ensuring an efficient field vaccination campaign, conducting appropriate surveillance that takes into account the risk of asymptomatic circulation, and monitoring these two coordinated programs. The means and the costs required are important and add up with the costs of the other essential prevention and control measures. Furthermore, constant monitoring and even vaccine evolution would be needed to ensure that vaccines remain protective against the circulating viral strains.

## 4. The Increasing Zoonotic Risk

### 4.1. Numerous Viral Reassortment Events

Monitoring of HPAI clade 2.3.4.4 b virus strains has revealed numerous viral reassortment events with other avian influenza viruses. From October 2022 to March 2023, 20 distinct genotypes were identified in Europe, four of which have been present since the 2021–2022 season, and 16 of which are the result of reassortment with other viruses present in Eurasian wild birds [5]. In France alone, nine different genotypes have been detected since October 2022, and 14 in total since 2021 [43,58]. The two genotypes most commonly found in France and Europe were: genotype AB (EURL nomenclature), associated with the majority of outbreaks in poultry and also very present in wild waterfowl, and genotype BB (EURL nomenclature), which mainly concerns wild shorebirds and is most likely the result of reassortment with a Laridae-associated low pathogenicity AI virus of H13 subtype.

The apparent high capacity of the clade 2.3.4.4b virus lineage to reassort with other influenza viruses increases its propensity to adapt to other bird species, but also to mammals. Figure 7 presents the major risks for human health of the reassortment of these viral strains with other influenza viruses in domestic animal populations. The main concern is the risk of reassortment of these strains with influenza viruses of swine or human origin via turkey and pig farms. Compared with other poultry species, turkeys are far more receptive to swine influenza viruses. They can also be infected by a human-origin viral influenza strain, virus A (H1N1) pdm09 [59,60,61]. Both avian and human influenza viruses can infect pigs. Moreover, the pig population is already affected by significant circulation of influenza virus strains, some of which are the result of reassortment between pig and human viruses [62].

### 4.2. Frequent Crossing of the Species Barrier

These HPAI strains are increasingly being detected worldwide following the crossing of the species barrier. A wide variety of terrestrial and marine mammals are involved. In all, at least 28 different species have been recorded, including the fox, mink, ferret, bear, lynx, badger, sea lion, seal, dolphin, cat, dog, and pig. For some species, these crossing events have been marked by massive mortality events (Figure 8). This was the case for seals and sea lions [63,64,65]. While considerable excess mortality could be the result of the consumption of dead or diseased birds, or of massive environmental contamination by infected birds (the areas concerned were also affected by excess mortality in wild birds), intraspecies circulation is suspected in a number of cases. This is notably the case for marine mammals [63] and it has been confirmed for an outbreak on a mink farm in Spain [66]. In France, three HPAI detections involved mammals: a domestic cat [67] with a strong link to an infected poultry farm, a captive bear in a zoo concerned by detections among captive and wild birds, and a wild fox in an area with massive Laridae excess mortality [11]. Furthermore, contamination within certain mammal species may be underestimated through current mortality-based surveillance programs, as shown by a study conducted on wild carnivores in 2020–2022 in the Netherlands [68]. Several markers known to be associated with increased pathogenicity or adaptation to mammalian hosts, including the PB2-E627K, -D701N, and -T271A mutations, were identified in strains that infected mammals [11].

### 4.3. Adapting HPAI Surveillance for Mammals

In response to the change in the risk of species barrier-crossing events, HPAI surveillance in France has been extended to wild mammals, as part of event-based surveillance in areas affected by episodes of mass excess mortality in wild birds. The SAGIR network and the French national network monitoring marine mammal stranding cases (Réseau National Echouages; RNE) [69]) were enrolled for this purpose. The ESA Platform’s dedicated group proposed a risk-based protocol for event-based surveillance of HPAI in wild mammals to competent authorities. This protocol targets the most at-risk species (large and meso wild carnivores, mustelids, and marine mammals) and areas affected by HPAI cases in wild birds.

On every mixed poultry/pig farm affected by HPAI and on pig farms very close to an HPAI poultry outbreak, mandatory active surveillance is conducted on the pigs [70]. In addition to this surveillance, reinforcement of event-based surveillance for the most at-risk domestic mammal has been implemented this last HPAI season, with messages directly targeting the people in charge of surveillance. To do this, the French Ministry of Agriculture and Food Safety (MASA) distributed an alert message about the HPAI case in a cat to veterinarians. In addition, the French National Surveillance Network for swine influenza A viruses (RESAVIP) [71] alerted its members about the risk of the introduction of avian influenza viruses on pig farms in order to enhance the reactivity of the network.

### 4.4. Adapting HPAI Surveillance and Prevention for Human Health

First, actions concerning human health have focused primarily on protecting people potentially directly exposed to avian influenza viruses. The MASA developed specific leaflets [72] for those involved: people collecting wild birds found dead and people exposed to infected birds or birds with suspected infection and their products (feathers, droppings, etc.). The recommendations focused on wearing appropriate personal protective equipment and on strict observance of biosafety measures to minimize the risk of virus dispersal. In addition, in order to limit the risk of reassortment between animal and human viruses, the French National Authority for Health (Haute Autorité de Santé; HAS) recommends vaccination against human seasonal influenza, and this vaccination is reimbursed for people potentially exposed in the course of their work (farmers, veterinarians, and technicians). Furthermore, people with influenza-like illness are advised to avoid all contact with birds at risk of avian influenza, in order to limit contact between human and avian viruses, and thus reduce the risk of reassortment.

Second, actions were focused on monitoring exposed human populations. Messages sent by both the MASA and the French national public health agency (Santé publique France; SpF) to these populations have strengthened existing passive surveillance: anyone exposed to HPAI viruses should report any flu-like symptoms to their physician [73]. At the same time, SpF has developed the SAGA program for active surveillance of human health around outbreaks of HPAI on livestock farms. This program will be implemented in the very near future.

## 5. Conclusions

Facing an increasingly threatening, urgent, and often surprising context, we have had to constantly adjust our preventive measures, as well as response and control, both during and between crises. In order to find the best solutions, we have questioned and will no doubt need to continue questioning some of our certainties and paradigms. We also need to develop our forecasting abilities, particularly with regard to the zoonotic risk represented by these HPAI virus strains: surveillance of exposed populations, prevention of exposure risks, specific risk-adapted methods and capacities adapted to this risk for decontamination, and depopulation of infected farms.

The control of HPAI on poultry farms must be based on the combined effect of: (i) biosecurity measures at all levels of the poultry industry, (ii) in-depth reorganization of poultry production throughout the country (lower density, adjustment of farm size to the HPAI risk, reduction in areas where highly receptive species and/or species that may contribute to zoonotic risk are concentrated, etc.), and (iii) risk-based surveillance. Vaccination of poultry species that contribute most to the risk of epizootic disease on farms will support these measures. This will hopefully lead to greater control of HPAI circulation in the domestic compartment.

The other challenge concerns the widespread circulation of these virus strains in wild birds, with episodes of significant excess mortality in certain species (raising conservation issues), but also very probably in a more silent way in others. To address this challenge, close and precise surveillance of the situation in wild birds will be essential to better understand and anticipate the risks involved. We will also have to imagine new solutions to protect endangered species from HPAI.

## Figures and Tables

**Figure 1 viruses-16-00101-f001:**
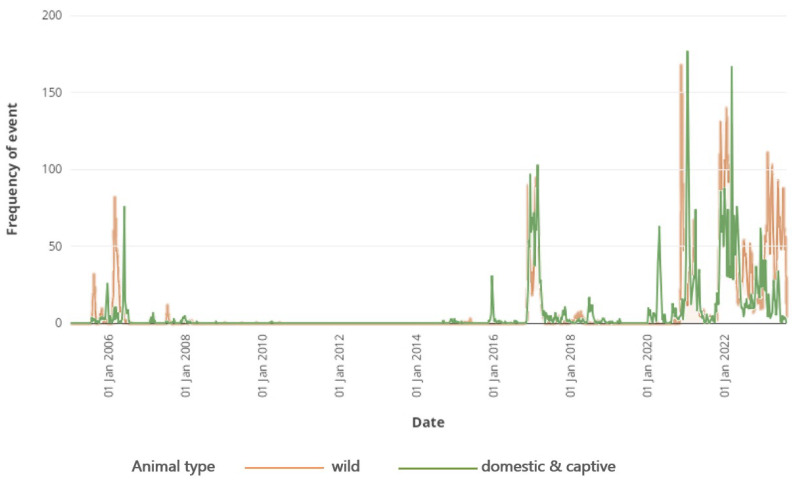
Distribution of the number of HPAI H5 virus detections in Europe per week from January 2005 to July 2023 (Source: FAO Empres-i, [10]).

**Figure 2 viruses-16-00101-f002:**
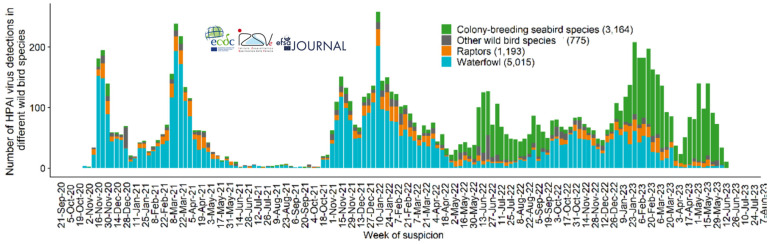
Distribution of total number of HPAI virus detections reported in Europe by week of suspicion (dates indicate the first day of the week) and affected wild bird categories (10,147), from October 2020 to 23 June 2023 (Graph source: EFSA Scientific Report, June 2023 [11]).

**Figure 3 viruses-16-00101-f003:**
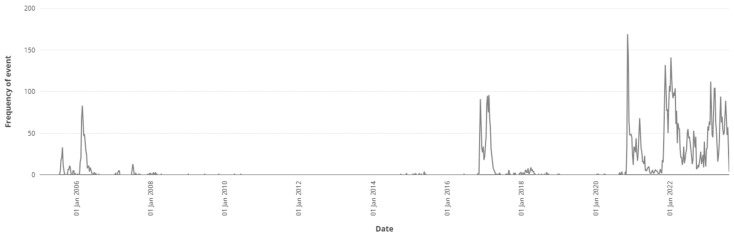
Distribution of the number of HPAI virus detections in wild birds reported in Europe per week from January 2006 to July 2023 (Source: FAO Empres-i [10]).

**Figure 4 viruses-16-00101-f004:**
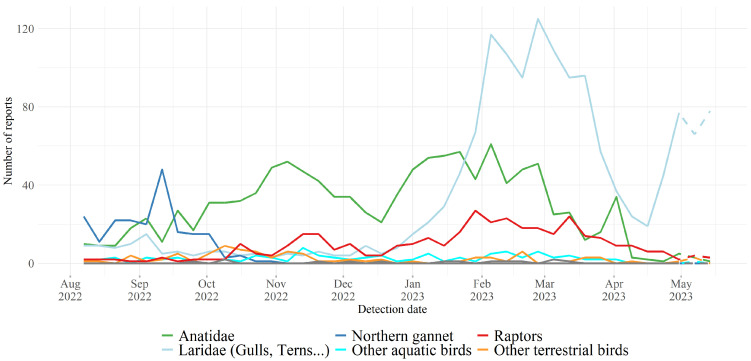
Weekly incidence (number of reports per week) of cases in wild birds by wild bird species group in Europe from 1 August 2022 to 22 May 2023 (Graph source: BH-VSI of the ESA Platform [22]; Data source: European Commission ADIS, WAHIS-OMSA).

**Figure 5 viruses-16-00101-f005:**
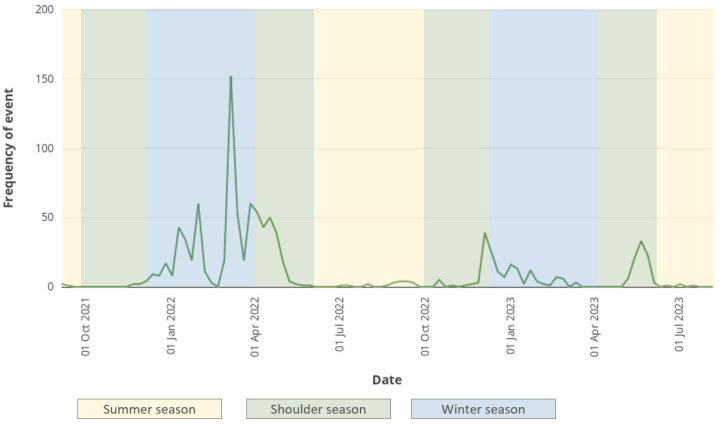
Distribution of the number of HPAI H5 virus detections in domestic poultry (flocks, backyard flocks) and captive birds reported in France per week from September 2021 to July 2023 (Source: FAO Empres-i [10]).

**Figure 7 viruses-16-00101-f007:**
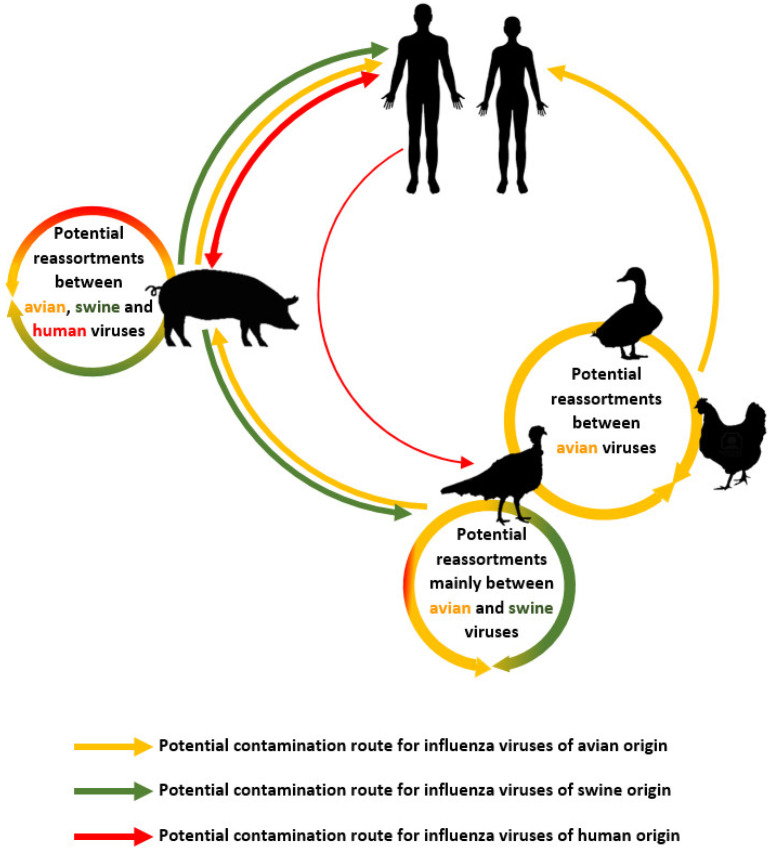
Simplified diagram of potential influenza virus reassortment risks for human health between avian, swine, and human influenza viruses in domestic animal populations.

**Figure 8 viruses-16-00101-f008:**
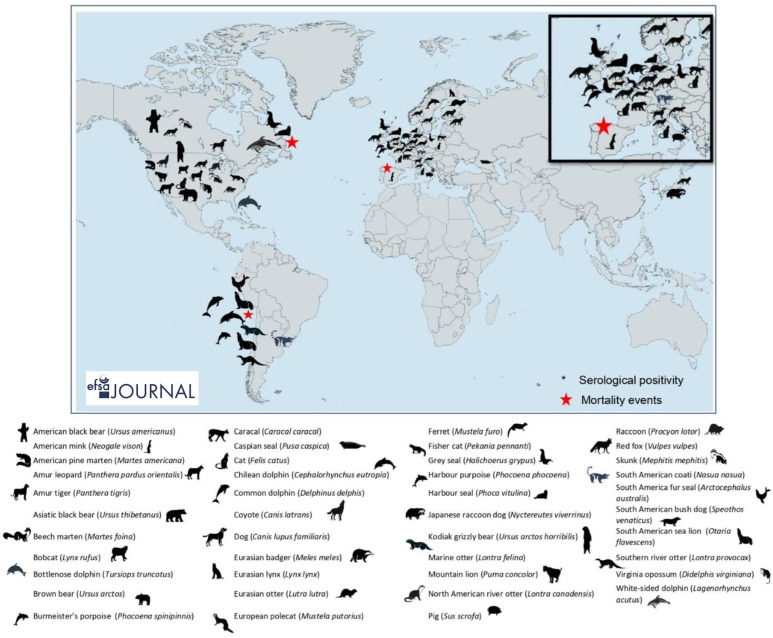
Geographical distribution by country and species of detections of HPAI in non-human mammals worldwide since 2016 (Graph source: EFSA scientific report [11]).

## Data Availability

No new data were created or analyzed in this study. Data sharing is not applicable to this article.

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
