# Peer review of "New Patterns for Highly Pathogenic Avian Influenza and Adjustment of Prevention, Control and Surveillance Strategies: The Example of France"

_viruses, 2024, doi:10.3390/v16010101_

Round 1

Reviewer 1 Report

Comments and Suggestions for Authors

Comments on the Quality of English Language

Author Response

Dear reviewer,

Thank you for your supporting review.

We have corrected the manuscript in accordance with your comments of editorial nature.

We hope this will answer your request adequately.

Axelle Scoizec

Reviewer 2 Report

Comments and Suggestions for Authors

The manuscript is a opinion paper.  The authors have done a nice job of presenting the current disease situation in France and around the globe.  Prevention and control strategies related to HPAIV are important and a single approach has not been accepted.  

I only have a few comments. 

Line 151: The birds are infected with the virus and develop disease thus, it should be HPAIV-infected birds not HPAI-infected birds. 

Line154: Adjusting HPAI virus surveillance in wild birds. Again, the surveillance efforts are targeting the presence of the virus not just the disease. 

Line 253: California condors have now been vaccinated to protect against infection by HPAIVs.  It is no longer being considered.....it has been done. This statement should be updated.

Lines 333-334: Why not propose new guidelines for laying hens and turkeys? These are some of the most at risk populations and the hardest to respond to. Were the guidelines not changed because they were already edited? Please explain.

Line 416: Vaccination can limit transmission.....or not. Was the study performed using homologous challenge strains or heterologous strains? If the field virus has changed from the seed used for vaccine production then vaccination may not help. Readers will likely not refer back to the original data. It should stated here. 

Lines 431-434: There was no indication that the farms in Figure 6 were infected by wild birds or horizontal transmission from infected farms. This may change the way you choose to write this closing remark. It is clear that the authors support the use of vaccines but fail to raise the proper questions surrounding their use and efficacy (or lack thereof).  The authors do state that cases are hard to detect in the wild animal population. Vaccination will make cases harder to detect in commercial settings once the main clinical disease sign is potentially removed.........death. The vaccines, if matched to the circulating virus, will decrease but not stop viral shedding and will also decrease mortality. 

We have adopted many management styles that include non-contained environments. This stand to improve animal welfare but also increase risk of exposure to not just HPAIV but many diseases. The choice to vaccinate is not as easy as presented. I wish this was a more balance report. It is listed as a Perspective so it is their right to present but a single viewpoint. 

Author Response

Dear reviewer,

Thank you very much for your detailed review of our manuscript.

We have corrected the document as requested for the lines: 151, 154.

For the next points, the changes made on the manuscript are highlighted in green in the new version of the manuscript.

We have updated the information and the reference about Condors vaccination as requested.

We have more detailed the adjustements made for poultry confinement for galliforms and clarified the exception for turkeys and hens.

For the vaccination experimentation, we have detailed the strain used for the challenge and the vaccines H5 sequences'origins  in the manuscript.

We have added references about the introductions of viruses in France.

We have amended the last pragraph of the section 3.4 in order to briefly expose the challenges and the difficulties represented by the vaccination strategy.

We hope that the changes answer adequately the request.

Best regards,

Axelle Scoizec

Reviewer 3 Report

Comments and Suggestions for Authors

Author Response

Dear reviewer,

Thank you for your careful review.

For the minor comments : we have corrected the line 18 as requested. For the Figure 8 unfortunately we cannot change the figure as requested because we have to reproduce it exactly as it is in the original paper.

For the major comments (changes are highlighted in green in the document):

  1. We have added (lines 96-105) a paragraph about the flyways of bird migrating to France.
  2. We have detailed the genotypes of HPAIV mostly encountered during the summer seasons on wild birds according to the species (lines 106-123).
  3. We have added the bird species families targeted by OFB for the surveillance and added examples to illustrate the system (lines 180-189)   

We hope the changes made in the manuscript will answer adequately the request.

Best regards,

Axelle Scoizec

Round 2

Reviewer 3 Report

Comments and Suggestions for Authors

The authors throughly addressed to the reviewers' comments.